# Nursing students' perceived anxiety and heart rate variability in mock skill competency assessment

**Cho Lee Wong****[1]\*, Wai Tong Chien**[1], **Mary Miu Yee Waye**[1], **Mark Wun Chung Szeto**[2], **Huiyuan Li**[1]

1 The Nethersole School of Nursing, Faculty of Medicine, The Chinese University of Hong Kong, Hong Kong, SAR, China, 2 School of Life Sciences, The Chinese University of Hong Kong, Hong Kong, SAR, China

\* jojowong@cuhk.edu.hk

## Abstract

### Background

Skill competency assessments induce stress and anxiety and may affect nursing student performance. Little is known about stress and perceived anxiety levels and their relationship in the mock skill competency assessment.

### Methods

A cross-sectional study was conducted to examine the stress levels (as assessed by heart rate variability, HRV) and perceived anxiety before, during and after the mock skill competency assessment, and to explore their relationships to performance in a total of ninety first-year undergraduate nursing students.

### Results

The HRV decreased significantly during the assessment and increased significantly 10 min after the assessment ($p < 0.01$). Higher performers showed significantly lower HRV during and after the assessment ($p < 0.01$). The assessment score was negatively correlated with HRV during and after the assessment ($p < 0.05$).

### Conclusions

Considering assessment-related stress and anxiety through a mock assessment prior to the actual skill assessment provides implications for future nursing education.

## Introduction

Admission into a professional degree programme such as nursing marks the beginning of achieving a career goal. The nursing curriculum aims to cultivate students with a strong theoretical and knowledge base in biological, socio-behavioural and nursing sciences [1]. In

**Data Availability Statement:** The datasets generated and analysed during the current study are not publicly available due to privacy protection and ethical restrictions by the Joint CUHK-NETEC

Clinical Research Ethics Committee. The Chinese University of Hong Kong's Research Ethics Board has imposed data sharing restrictions as they do not allow identifiable or de-identified data to be electronically transmitted outside of university including open access repositories. Requests can be sent to Clinical Research Ethics Office (Phone: 852 3505 3935; email: crec@cuhk.edu.hk). Requests can also be sent to the corresponding author (jojowong@cuhk.edu.hk).

**Funding:** Innovative and Technology Fund for the Public Sector Trial Scheme funded CLW (project reference: SCT/003/20SP). https://www.itf.gov.hk/en/funding-programmes/facilitating-technology/psts/index.html The funders did not play any role in the study design, data collection and analysis, decision to publish, or preparation of the manuscript.

**Competing interests:** The authors have declared that no competing interests exist.

**Abbreviations:** ACLS, Advanced Cardiovascular Life Support; C-STAI, Chinese version of STAI; CUHK-NTEC, Chinese University of Hong Kong-New Territories East C; HR, heart rate; HRV, heart rate variability; ms, milliseconds; OSCE, Organization for Security and Co-operation in Europe; PPG, photoplethysmography; RA, research assistant; R-R, rhythm-to-rhythm; SD, standard deviations; SF-STAI, Short-Form State-Trait Anxiety Inventory; STAI, State-Trait Anxiety Inventory.

addition, it emphasises the ability of students to make critical, informed judgements and reasonable decisions in nursing practice, as well as the skills competency to provide patient care [2]. In order to ensure that students can provide safe and high-quality patient care in clinical environments, students are being taught the essential components and psychomotor skills for performing the nursing skills and then being regularly assessed in a set laboratory environment to ensure that the required competency level is achieved before implementing in clinical areas.

Although skill competency assessment is an indispensable part of nursing education and clinical practicum, the frequent and demanding skills assessments are considered stressful and anxiety-provoking for most nursing students [3–5]. Stress responses arise from any physical or psychological stimuli that disrupt homeostasis [6]. Anxiety, one the other hand, is a mood state characterised by autonomic responses that trigger vague and uneasy feelings, causing the individual to experience or anticipate threatening or dangerous events [7]. Nursing students often experience significant degrees of stress and anxiety due to frequent testing of fundamental nursing skills, heavy assignments in various clinical settings, fear of making mistakes, concerns about the pass/fail grade, interactions with clinical staff and faculty, and deficiencies in professional nursing knowledge and skills [8]. Previous studies have shown that the stress and anxiety levels experienced by nursing students were higher than that experienced by students of other healthcare majors [5, 9, 10]. An integrative review concluded that the key sources of stress amongst undergraduate nursing and midwifery students emanated from clinical and academic issues [11]. In particular, preparation for and sitting examinations, coupled with the fear of failure and making mistakes, were identified as the main assessment-related stressors experienced by clinical nursing students [12, 13]. Although medical students perceive stress within an acceptable range as beneficial to their academic work [14], excessive stress and anxiety levels can negatively impact nursing students' overall assessment performance [4, 15]. Moreover, Khalaila [16] found that test anxiety (refers to responses that accompany concern about possible negative consequences of failure on an exam) [17], negatively correlated with academic achievement in a sample of undergraduate nursing students. More recently, Brodersen and Lorenz [18] revealed that nursing students who failed an exit exam experienced significantly higher levels of perceived stress after the exam than those who passed the exam. As such, nurse educators need to understand students' anxiety and stress levels in response to assessments. These allow them to provide students with more effective support and interventions to respond flexibly to stress during their undergraduate education, thereby improving academic performance.

To date, little is known about nursing students' stress and anxiety levels during the skill competency assessment. However, studies have consistently shown that during written or clinical examinations, university students' self-reported stress and anxiety are generally increased [4, 18–21]. Compared to a low-stakes homework condition (part of the prelicensure course requirement), prelicensure nursing students experienced higher perceived stress and a higher level of salivary alpha-amylase after a high-stakes exit examination (i.e. part of the curriculum of the prelicensure nursing programme) [18]. Specifically, Ping et al. [21] examined the patterns of anxiety symptoms amongst medical students during a clinical examination assessed using a self-reported Spielberger Test Anxiety Inventory-Trait. The results suggested that perceived anxiety symptoms peaked around 10 minutes before the assessment but decreased as the assessment progressed [21]. Additionally, evidence suggests that female students are more likely to perceive stress on academic performance than male students [22]. Sex is known to play an important role in human development and behaviours, with potential cardioprotective effects of estrogen in females [23]. As such, there may be sex differences in HRV. Studies have also shown that sex differences exist in study habits and coping mechanisms to stress, which can significantly impact academic performance [22].

Some studies have adopted heart rate variability (HRV) as a reliable measure of the sympathetic and parasympathetic activity of the autonomous nervous system to assess the stress levels of students, instead of relying on self-reported questionnaires [24, 25]. HRV is also an important biomarker for capturing individuals' psychological status, such as anxiety [26, 27]. HRV is described as the variation over time of the intervals (i.e. rhythm-to-rhythm (R–R) interval) between consecutive heartbeats and oscillations between consecutive, instantaneous heart rates [28]. If a person is under stress, the sympathetic activity will increase, and at the same time, the vagal activity of the heart will decrease, increasing the heart rate (HR) with a decrease in HRV [29]. Several studies have found that when compared with non-academic occasions such as during the semester [30], during holidays [25] and after holidays [31], university students exhibited lower HRV before or during examinations. Hammoud et al. [32] investigated the changes in HRV in a group of 90 science students 1 hour before, 2 hours during, and 1 hour after a final examination. Results suggested that HRV after the examination was significantly higher than that before or during the examination, indicating a release from examination-related stress after the examination. Additionally, male students presented significantly higher HRV than female students before and after the examination, suggesting a sex difference in stress management [32]. To date, HRV is rarely used in nursing programmes as a critical physiological measure of students' anxiety. An optimal level of HRV reflects healthy function and an inherent self-regulatory capacity, adaptability, or resilience [33]. Therefore, incorporating HRV as an objective measure of nursing students' stress during assessments may promote the use of physiological measurement technologies in nursing education, as well as help assess students' adaptability to stress.

For high-stakes tests (e.g. skill competency assessments), in which students would face significant consequences such as failing to enrol in a particular course or having to remain at a certain level until the requisite score is achieved, may make their experience more stressful [34]. Mock skill competency assessment is therefore adopted as a common method by faculty. The design is structured as an additional step in the standard competency assessment so that students, prior to undergoing the faculty-led competency assessments, are evaluated for their performance in a simulated competency assessment [35]. This can allow students to realistically simulate the actual assessment at the time, so that students can be fully familiar with the assessment process and be prepared in advance for the potential challenges and difficulties that may be faced in the actual assessment. Meanwhile, it also allows educators to identify in advance those underperforming with unusually high stress levels to provide them with support to improve student performance and relieve their high stress levels. Extensive evidence has demonstrated that HRV is known to relate to an emotional condition and a stressful situation, and to correlate with mental and psychological states after stress due to various academic examinations in students [14, 36, 37]. However, evaluation of the physiological and psychological anxiety and performance in mock skill competency assessment prior to the actual assessment, and their associations throughout the mock skill competency assessment are limited. Monitoring physiological HRV-based stress in undergraduate nursing students before the actual assessment may help build evidence on new ways to evaluate students' performance and overall professional qualities in nursing programmes.

The overarching aims of this study were (1) to examine the perceived anxiety and HRV and their changes in nursing students across (before, during and after) the mock skill competency assessment; (2) to examine the differences in perceived anxiety, HRV and performance between female and male students across (before, during and after) the mock skill competency assessment; (3) to examine the relationships amongst perceived anxiety, HRV and performance in nursing students.

## Materials and methods

### Design

This study adopted a cross-sectional correlational design.

### Setting and participants

This study was conducted at the Nethersole School of Nursing, Faculty of Medicine, The Chinese University of Hong Kong. Two hundred and sixteen first-year Bachelor of Nursing students enrolled in a mandatory fundamental nursing course were informed about the study and invited to participate. Those taking any medication or having health conditions that could induce abnormal changes in the heart rate signal and those with a history of anxiety and stress disorders were excluded.

### Course and mock skill competency assessment

The fundamental nursing course is a first-year undergraduate-level course that commences in the second term of the academic year. The course encourages students to understand the application of basic principles of nursing care to help clients meet their basic living needs safely. Students must attend a 2-hour lecture and a 2-hour practical session every week. The final assessment includes a 2-hour written examination and a skill competency examination. The skill competency assessment is designed to enable students to demonstrate their level of competency in aseptic technique using a clinical scenario in a structured laboratory setting. Students are expected to perform the scenario about wound dressing on the manikin by adopting the nursing process, including assessment, planning, implementation, evaluation and reporting [38]. Passing the course is a prerequisite for subsequent clinical placement. Since this is the first skill competency assessment, a mock skill competency assessment is arranged before the actual assessment to help students familiarise themselves with it [35]. The perceived anxiety, HRV and performance of nursing students in this mock skill competency assessment were examined in this study.

### Outcome measures

**Socio-demographic characteristics.** Socio-demographic data of students were collected, including gender, age, smoking status (yes/no), alcohol drinking status, caffeine consumption, physical exercises, periodic medication and medication taken in the last 24 hours.

**State-Trait Anxiety Inventory (STAI).** The State-Trait Anxiety Inventory (STAI) was developed by Spielberger et al. [39] to measure state and trait anxiety. The STAI consists of 40 items clustered into two subscales: State-Anxiety (20 items) and Trait-Anxiety (20 items) subscales. Items in the STAI are measured on a 4-point scale (from 1 to 4). The subscale scores are summed to give a total subscale score ranging from 20 to 80, with higher scores indicating higher anxiety levels [39]. Tsoi et al. [40] translated the Chinese version of STAI, and the internal consistency of C-STAI was 0.90 [41]. As time is a consideration for students to fill in STAI before their mock skill competency assessment, the Short-Form State-Trait Anxiety Inventory (SF-STAI) [42], a six-item version of the Spielberger STAI, was adopted in this study. Participants were suggested to indicate the extent to which they currently feel comfortable, anguished, eased, nervous, concerned, and good. The items concerning comfortable, eased, and good are reverse scored. The scores are summed to give a total score ranging from 6 to 24, with greater scores indicating a high level of anxiety. The Cronbach's alpha coefficient of the current study was 0.89.

**Heart rate variability.** The stress level amongst participants was objectively assessed by a wearable earphone device using ActivHeartsTM dynamic heart rate sensing technology. This earphone device uses photoplethysmography (PPG) to provide accurate heart rate and HRV measurements in this study. PPG is a non-invasive technology that uses a light source and a photodetector at the skin's surface to measure the volumetric variations of blood circulation, which is unlikely to distract the participants' daily activities [43]. The popularity of PPG technology as an alternative heart rate monitoring technique has recently increased, mainly due to the simplicity of its operation, the wearing comfortability for its users, and its cost-effectiveness [44]. Before the assessment, students were asked to wear earphones. The HRV could be directly recorded using the ActivHeartsTM dynamic heart rate sensing technology in the earphone device according to the guidelines of the task force of the European Society of Cardiology and the North American Society of Pacing and Electrophysiology [45]. The R–R interval reduction represents the degree of stress at a particular moment, and the frequency of heart rate peaks represents the number of episodes of elevated stress [30].

**Performance during mock skill competency assessment.** The student performance during the mock skill competency assessment was evaluated by the scores obtained in the mock assessment rated by a nursing tutor/educator. The competency form consists of 19 items assessing three aspects of skills performance by individual students, including assessment and planning skills (6 items), implementation skills (12 items), and evaluation skills (2 items) on performing a wound dressing. Items are measured on a 6-point Likert scale from 0, "not done", to 5, "very good". Each of these three sets of skills would contribute to a different percentage of the total score. The total score of the assessment is 0 to 100, with a passing grade of 50. The RR interval relevant to skills performance was grouped and categorised into three levels in terms of the range of their assessment scores, including low (score below 68; rate as C or below), moderate (score between 68 to 79; rate as B- to B+), and high (score 80 or above; rate as A- or above) performers. The levels and score ranges are identified by the school's programme director, course coordinator and education committee. It has been widely used in various nursing courses in nursing school and as an evaluation standard for nursing students' performance.

## Procedure

All first-year undergraduate students enrolled on the course were approached by a research assistant (RA) after a lecture one month before the mock skill competency assessment. The RA introduced and explained the purpose and procedure of the study to the respective students in the lecture hall. Students were provided with an information sheet detailing the study objectives. Those interested in the study were invited to sign a written informed consent form.

Individual participants were required to download a mobile application on respective application stores (Google Play and Apple iTunes). The main functions of the mobile application included: (1) inputting questionnaire responses and (2) uploading the HRV data. All participants received a briefing on how to download the mobile application, use and recharge the earphone device, and upload the HRV data.

Each participant was asked to abstain from any caffeinated beverages/alcohol/smoking on the day of the skill competency assessment. They were required to arrive at the assessment venue 30 minutes before the assessment. Upon arrival, each participant was asked to wear the earphone device and connect them to the mobile application, through which they entered their responses to questionnaires, including STAI-State 10 minutes before the assessment. Afterwards, they were asked to sit in a chair and breathe regularly without talking for 20 minutes. HRV was measured over 10 minutes before, throughout the 20-minute mock skills

competency assessment, and 10 minutes after the assessment. Ten minutes after the assessment, students were asked to complete the questionnaires again in a quiet room next to the assessment venue. The mock skill competency assessment performance was obtained from the assessment records. Participants who completed all data collection received a $100 coupon as an incentive.

## Ethical considerations

Ethical approval was obtained from the Joint CUHK-NTEC Clinical Research Ethics Committee (Ref. No. 2020.502). All study procedures involving human participants complied with the Declaration of Helsinki. Students were assured that their participation was entirely voluntary and their right to withdraw at any time was upheld. Each participant was assigned a unique study ID, and the downloaded questionnaire and HRV data were compiled in an encrypted data file without any personal identifying information. All collected information was kept confidential, and the encrypted data file was only accessible to the research team. All electronic data would be destroyed three years after the completion of the study.

## Data analyses

All statistical analyses were conducted using IBM SPSS Statistics (version 25.0, IBM Corporation, New York, USA). Questionnaire responses and HRV data were downloaded, coded and input into the SPSS. HRV data were generated directly by ActivHeartsTM dynamic heart rate sensing technology in the earphone device and uploaded into the mobile application. Raw data was inspected for any artefacts [46]. The HRV data for each student were divided into three segments: 10 minutes prior to the assessment, 20 minutes during the assessment, and 10 minutes after the assessment. Descriptive statistics, including means, standard deviations (SD), medians, interquartile range (P25, P75), frequencies, and percentages, were used to summarise and present students' characteristics and all outcome measures.

Since the data were not normally distributed, nonparametric tests, including the Wilcoxon Signed Rank test, were used to assess changes in perceived anxiety scores measured at two points within the group. Friedman test was adopted to test the changes in HRV as measured by RR interval (milliseconds, ms) on three occasions. Kruskal-Wallis test was used to compare the differences in HRV in three levels of performers. Post-hoc tests with all pairwise comparisons were further conducted to explore differences between any two groups if a significant difference was indicated in three groups. Mann-Wahitney U test was used to compare scores on continuous variables for two unpaired groups, especially exploring differences in median RR interval [ms], STAI, and scores during mock skill competency assessment between male and female nursing students. Finally, the Spearman test was used to evaluate the correlations between two continuous variables, such as the RR interval with perceived anxiety and assessment scores. The statistical significance of all tests was set at $p < 0.05$ (two-sided).

## Results

### Participants' characteristics

A total of ninety students participated in the study, with a response rate of 41.7%. Participants were 18.40 (SD = 0.80) years, with the majority females (74.4%). All of them had never smoked, and 95.6% (n = 86) were not on periodic medication. Most did not meet the World Health Organization's physical activity guideline (82.2%), which is at least 150 minutes of moderate-intensity aerobic physical activity throughout the week. No significant differences

**Table 1. Outcome measures across time.**

| | All participants (n = 90) | Female (n = 67) | Male (n = 23) | Z | p-value |
|---|---|---|---|---|---|
| *Perceived anxiety scores* | | | | | |
| T0 | 11.00 (9.00, 15.00) | 11.00 (9.00, 15.00) | 12.50 (9.75, 17.00) | -1.074 | 0.27[a] |
| T2 | 11.00 (8.50, 15.00) | 11.00 (9.00, 15.00) | 12.50 (7.75, 14.25) | -0.242 | 0.81[a] |
| Z | -0.454 | -0.259 | -0.451 | | |
| p-value | 0.65[b] | 0.80[b] | 0.65[b] | | |
| *HRV (measured by mean RR interval[ms])* | | | | | |
| T0 | 677.54 (616.48, 753.21) | 679.62 (620.19, 754.67) | 666.75 (583.64, 750.44) | -1.013 | 0.31[a] |
| T1 | 524.94 (474.61, 588.45) | 522.74 (473.14, 587.68) | 544.44 (484.30, 598.46) | -0.708 | 0.48[a] |
| T2 | 704.26 (637.48, 780.76) | 711.61 (642.36, 789.05) | 693.52 (599.75, 766.37) | -0.733 | 0.46[a] |
| $\chi^2$ | 101.802 | 81.300 | 21.238 | | |
| p-value | < 0.01[c] | < 0.01[c] | < 0.01[c] | | |
| *Skill competency assessment scores* | 73.08 (63.20, 79.20) | 72.35 (63.53, 79.05) | 75.00 (62.20, 80.20) | -0.985 | 0.33[a] |
| High* | 20 (22.22) | 14 (70.00) | 6 (30.00) | -0.206 | 0.84[a] |
| Medium* | 44 (48.89) | 33 (75.00) | 11 (25.00) | -1.057 | 0.29[a] |
| Low* | 26 (28.29) | 20 (76.92) | 6 (23.08) | -1.156 | 0.25[a] |

*Data were presented as number (percentage) and others as median (P25, P75).

Abbreviations: ms, milliseconds; RR, rhythm-to-rhythm; T0, 10 minutes before assessment; T1, during the assessment; T2, 10 minutes after assessment.

[a] Mann-Whitney U test was used to compare differences between female and male students.

[b] Wilcoxon Signed Rank test.

[c] Friedman test

were indicated in the scores of the participants' socio-demographic characteristics between the male and female students ($p > 0.05$).

## Perceived anxiety before and after assessment

The scores of perceived anxiety 10 minutes before and 10 minutes after the assessment were 11.00 (9.00, 15.00) and 11.00 (8.50, 15.00), respectively. No significant differences were noted before and after the assessment and across gender (Table 1 and Fig 1).

## HRV before, during and after the assessment

The median HRV as measured by RR interval at 10 minutes before, during and 10 minutes after assessment were 677.54[ms] (616.48, 753.21), 524.94[ms] (474.61, 588.45), and 704.26[ms]

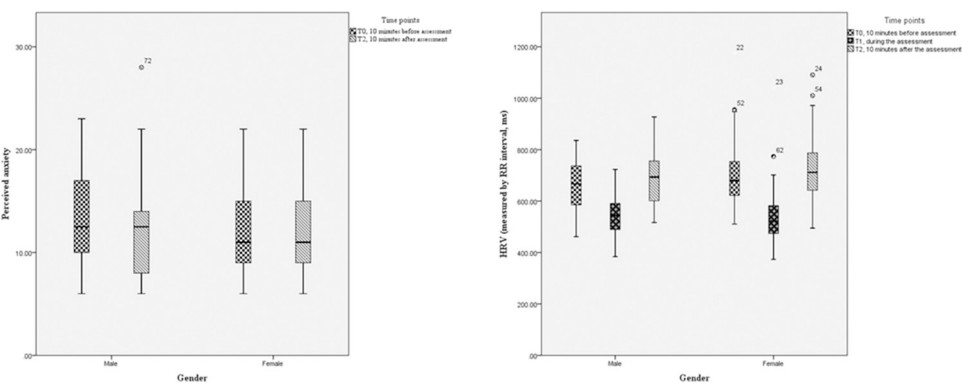

**Fig 1. Changes of perceived anxiety and HRV comparison between male and female students throughout the mock skill assessment.**

(637.48, 780.76), respectively. The median RR interval varied significantly across the three occasions ($p < 0.01$). The RR interval decreased significantly during the assessment, indicating that students were physiologically anxious during the assessment. The RR interval 10 minutes after the assessment increased compared to that before, indicating that the students were more physiologically relaxed after the assessment. There was no significant difference in HRV data between female and male students on the three occasions ($p > 0.05$, Table 1, Fig 1).

However, the Friedman test indicated significant within-group differences from 10 minutes before the assessment to 10 minutes after the assessment in HRV data ($p < 0.01$, Table 1). Post-hoc tests also (S1 Table and S1 Fig) showed that both male and female students showed significant decreases in their HRV from 10 minutes before the assessment to during the assessment, indicating significant increases in physiological stress (all $p < 0.01$). In addition, both groups showed significant increases in their HRV from during the assessment to 10 minutes after the assessment, indicating significant releases in physiological stress (all $p < 0.01$).

## HRV and mock skill competency assessment scores

Participating students' mean mock skill competency assessment scores were 73.08 (63.20, 79.20), indicating an overall average performance (Table 1). The percentages of the participants in each category were also summarised in Table 1. There was no significant difference between female and male students.

Table 2 and Fig 2 show the median HRV and perceived anxiety of the high, medium and low performers 10 minutes before, during and 10 minutes after the assessments. The results indicated that high performers were physiologically more stressed during and after the assessment. There were significant differences between the mean HRV during ($p < 0.01$) and after ($p = 0.04$) the assessment across the three groups. The post-hoc comparisons (S2 Table and S2 Fig) further showed that the HRV of medium ($p = 0.01$) and high performers ($p < 0.01$) were significantly lower than that of low performers during the assessment. The HRV of high performers were also significantly lower than that of low performers 10 minutes after the assessment ($p = 0.03$). In addition, the Friedman test indicated significant differences within

**Table 2. HRV of high, medium and low performers.**

|  | High performer (n = 20) | Medium performer (n = 44) | Low performer (n = 26) | $H$ | p-value |
|---|---|---|---|---|---|
| *Perceived anxiety scores* |  |  |  |  |  |
| T0 | 10.00 (7.25, 14.00) | 12.50 (9.00, 15.25) | 11.00 (8.00, 15.50) | 3.115 | 0.21[a] |
| T2 | 10.50 (8.75, 15.50) | 11.00 (8.00, 17.00) | 13.00 (8.50, 14.00) | 0.366 | 0.83[a] |
| *Z* | -0.024 | -0.843 | -0.104 |  |  |
| p-value | 0.51[b] | 0.38[b] | 0.48[b] |  |  |
| *HRV (measured by mean RR interval ᵐˢ)* |  |  |  |  |  |
| T0 | 654.96 (602.51, 748.52) | 682.59 (623.37, 746.35) | 679.03 (611.17, 825.45) | 1.013 | 0.60[a] |
| T1 | 505.79 (452.85, 563.67) | 510.12 (467.50, 548.70) | 573.05 (520.91, 667.81) | 11.795 | < 0.01[a] |
| T2 | 649.69 (588.89, 766.76) | 704.26 (647.97, 755.97) | 759.77 (658.83, 866.66) | 6.407 | 0.04[a] |
| $\chi^2$ | 27.100 | 52.769 | 22.455 |  |  |
| p-value | < 0.01[c] | < 0.01[c] | < 0.01[c] |  |  |

Data were presented as median (P25, P75).

Abbreviations: ms, milliseconds; RR, rhythm-to-rhythm; T0, 10 minutes before assessment; T1, during the assessment; T2, 10 minutes after assessment.

[a] Kruskal Wallis H Test was used to compare differences between high, medium and low performers.

[b] Wilcoxon Signed Rank test.

[c] Friedman test

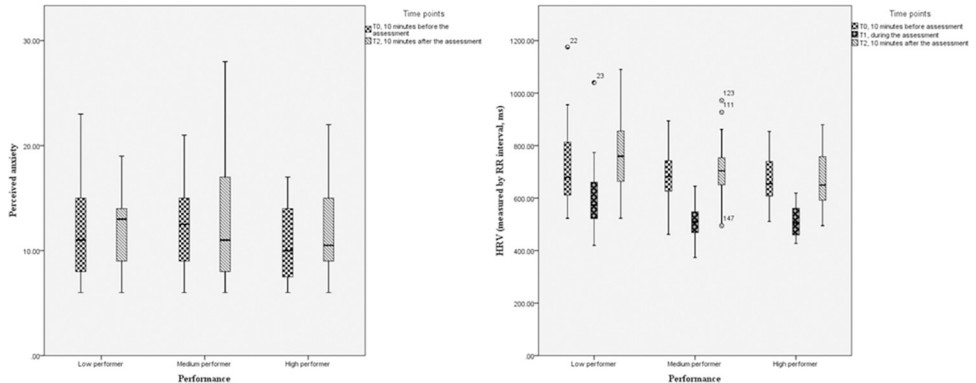

**Fig 2. Changes of perceived anxiety and HRV comparison in three performance-level groups during the assessment.**

three-level-performance groups in HRV, which revealed an average HRV trend that decreased first and then increased throughout the assessment process (Table 2 and Fig 2). The post-hoc tests further demonstrated this significant changing process (S3 Table and S3 Fig).

With regard to the score of perceived anxiety, non-significant differences were noted amongst the three groups before and after the assessment ($p > 0.05$). Within-group analysis also indicated non-significant differences in perceived anxiety scores at two time points within three levels of performers ($p > 0.05$, Table 2).

## Correlations amongst perceived anxiety, HRV, and mock skill competency assessment scores

Table 3 presents the correlations amongst the measures of perceived anxiety, HRV and mock skill competency assessment scores. The results showed a significant positive correlation between the scores of perceived anxiety 10 minutes before and 10 minutes after the assessment ($r = 0.41$, $p < 0.01$). The median HRV 10 minutes before the assessment showed significant negative associations with perceived anxiety 10 minutes before ($r = -0.37$, $p < 0.01$) and 10 minutes after the assessment ($r = -0.29$, $p < 0.01$), respectively. As shown in Table 3, the level of median HRV (RR interval) during the assessment was positively associated with that

**Table 3. Correlations between perceived anxiety, HRV and mock skill competency assessment scores.**

| | Perceived anxiety | | HRV (measured by mean RR interval) | | | Performance score |
|---|---|---|---|---|---|---|
| | Before [a] | After [b] | Before | During [c] | After | |
| **Perceived anxiety** | | | | | | |
| Before | - | 0.41** | -0.37** | -0.20 | -0.10 | -0.16 |
| After | - | - | -0.29** | -0.17 | -0.13 | -0.01 |
| **HRV** | | | | | | |
| Before | | | - | 0.68** | 0.46** | -0.20 |
| During | | | - | - | 0.40** | -0.38** |
| After | | | - | - | - | -0.32** |

** $p < 0.01$

[a] 10 minutes before the assessment

[b] 10 minutes after the assessment

[c] During the assessment

measured 10 minutes before the assessment ($r = 0.68$, $p < 0.01$). The median HRV (RR interval) 10 minutes after the assessment was also positively correlated with that measured at the other two time points ($p < 0.01$). Moreover, significant negative correlations were indicated between the scores of skills competency assessment and median HRV during ($r = -0.38$, $p < 0.01$) and 10 minutes after the assessment ($r = -0.32$, $p < 0.01$), respectively.

## Discussion

This study used HRV to objectively evaluate the physiological stress and self-reported questionnaire to evaluate the perceived anxiety of nursing students throughout the mock skill competency assessment. The results showed that HRV varied significantly before, during and after the assessment; in particular, the HRV decreased significantly during the assessment, indicating that participants were particularly stressed during this period. In addition, high performers had significantly lower HRV during and after the assessment than moderate and low performers. Results also showed that assessment scores were negatively correlated with HRV during and after the assessment, suggesting that some stress may be beneficial to assess performance.

Significant changes in HRV indicated a first decline and then an increase following the mock skill competency assessment for all participants. The changes in HRV concurred with those reported in Al-Ghareeb et al. [47], which found that students were physiologically stressed (as indicated by reduced variability in RR interval) during the assessment but felt relaxed at the end (as characterised by gradually increased variability in RR interval). A previous study has found that HRV can reflect an individual's capacity to self-regulate and adapt effectively to changing social or environmental demands [48]. The decline in students' HRV during the assessment indicated they had inherent self-regulatory adaptability to different nursing challenges or stressors, such as the nursing skills competency assessment. At the same time, students' knowledge and experience may influence the performance of nursing skills in a structured laboratory setting [49]. During their first year, entrance into an undergraduate nursing programme is a significant transition from the classroom to the clinical setting, fraught with uncertainty, ambiguity, and confusion [50–52]. First-year nursing students are engaged in challenging nursing coursework that involves both theoretical knowledge and practical skills. However, they may lack preparation and support when it comes to their first clinical skill operation. Since the aseptic technique is a relatively complex nursing skill for first-year undergraduates, students may not be sure what to expect or whether they have enough knowledge and relevant experience to manage the skills [47], resulting in higher stress and lower HRV during the assessment. In contrast, an increased post-assessment HRV might indicate that the students were becoming more familiar with procedures and assessment scenarios, which may help relieve their physiological stress, leading to higher post-assessment HRV.

When we compared the mean RR interval of high, medium and low performers, no significant difference was reported 10 minutes before the assessment, suggesting that the three groups experienced similar levels of anticipatory stress, with results comparable to a previous study [21]. The possible reason may be while mock skill assessment can evoke anticipatory stress, the relationship between anticipatory stress and performance likely involves the interplay of intrinsic factors, such as one's level of experience and perception of task-related demands versus resources, as well as extrinsic factors such as the complexity of the task and the nature and intensity of related stressors [53]. However, significant differences were observed between the three groups during and 10 minutes after the assessment, with high performers having significantly lower median HRV than moderate and low performers. Recent studies using self-reported measures to assess the relationship between stress and academic performance have yielded inconsistent findings, with some studies suggesting that higher

levels of stress during the examination period was associated with poorer average performance [54–56]. A study of medical students in their first clinical year even found a weak correlation between psychological distress and academic performance [36]. This difference might be because the characteristics of research participants were different, and the researchers relied heavily on self-reported questionnaires as a tool for measuring stress, which is thought to be not objective and scientific measurements for use as concrete guidance in actual educational situations. However, our results mirrored a recent study assessing the association between HRV-measures stress and written/skills examination scores. The results showed that students who experience higher stress with a lower level of HRV have higher academic performance [14]. The result likely because when students' stress increases to an optimal level, they may become more focused on their work and thereby improve their performance. This finding could be supported by the evidence of 'learning zones' that a moderate stress level benefits effective learning [57]. Nevertheless, the optimal stress level for achieving the best performance was largely unknown. On the other hand, it is suggested that an individual's adaptive success is based on the ability of the autonomic nervous system to cope with challenges [58]. Testing HRV can help confirm the flexibility or adaptability of the autonomic nervous system activity and assess the sympathetic nervous system/parasympathetic nervous system balance level to objectively evaluate the psychological and emotional states of an individual. Students with stronger autonomic nervous system responses, indicated by greater reductions in HRV and parasympathetic activity in response to assessments, and greater autonomic flexibility, tend to perform better [59]. When stress lasts for a certain period of time, the principle of homeostasis helps maintain balance in the mind and body, making stress a driving force that can increase an individual's efficiency and productivity [14]. However, students with poor coping skills may experience decreased and imbalanced autonomic nervous activity, leading to anxiety, depression, and poor academic performance. Future studies are needed to explore the optimal stress level that leads to the best performance and the association of autonomic flexibility with assessed performance and its associated factors.

Our results differed from those reported by an Australian group that studied second- and third-year nursing undergraduates enrolled in a two-hour simulation session [47]. Their findings suggested that a low level of anxiety might lead to optimal performance; by contrast, our study showed that high performers had high levels of physiological stress throughout the assessment. The difference could be explained by different analytical methods, in which their study categorised students according to their physiological anxiety levels. Our study categorised students into three performance levels (low, medium and high) and assessed the students' stress levels longitudinally. Nonetheless, our findings also highlighted a short-term dynamic changing trajectory in that three distinct groups of performers reflect a changing curve of HRV-measured stress, respectively. The results may help predict the variation of physiological stress throughout the mock assessment in nursing first-year students and to provide students with effective supportive and educational programmes for stress management based on HRV results during nursing education. Variations in HRV can be further explored to investigate individual differences in academic performance and clinical practice, which may help improve students' future performance in nursing or other health disciplines.

Our results demonstrated that mock skill competency assessment scores were inversely correlated with HRV during and 10 minutes after skills assessment, suggesting that stress might benefit the assessment performance. A previous study yielded similar findings, showing that simulated scenarios induced anxiety, leading to better performance and retention of ACLS skills in medical students [60]. Skills competency assessments require a certain degree of engagement and stress. Appropriate stress could increase an individual's attention to engagement, memory, and cognition abilities to adapt to changes in the external environment, which

could be seen as a positive psychological response [61]. Nevertheless, there was a lack of significant correlations between perceived anxiety levels and HRV during and after assessment and between assessment scores. A systematic review also found that OSCE-associated test anxiety has little effect on OSCE performance amongst health professional students [62]. However, it is notable that unlike HRV, which objectively assesses students' stress levels, perceived anxiety is a self-reported measure that may be influenced by psychosocial factors such as personality traits [14, 63]. The Hawthorne effect is also suggested as an influencing factor on the perceived anxiety level. Participants may modify their behaviour when they feel singled out for special attention [14, 63]. This possible reason may help explain why there might be differences in self-reported anxiety and HRV.

Interestingly, female students experienced lower levels of perceived anxiety and HRV-measured stress than male students before and after the assessment but showed higher levels of HRV-measured stress than male students during the assessment, despite the statistical insignificance reported. Further research is needed to investigate the influence of sex on professional performance among first-year nursing students.

## Limitations

Several limitations were noted in this study. First, HRV was assessed on the day of the skills competency assessment, but the stress level might reflect their stress on daily life activities and general health. Nevertheless, it has been shown HRV had satisfactory reliability in short-term recordings. Therefore the results likely reflect the students' actual stress levels due to mock skills competency assessments [45]. Second, students were only recruited from the first year of one cohort in the undergraduate nursing programme. Female students made up the majority of the sample, thus limiting the generalisability of the results. Third, this study revealed physiological stress and one psychological state (anxiety) and its correlations with mock assessment performance before an actual skills competency assessment. The results of this study might not be generalised to the physio-psychological condition of the nursing students during the actual skills competency assessment. Therefore, further investigation is suggested to examine nursing students' perceived anxiety, HRV and competency assessment scores during the actual skills assessment, compare the changes in the students' conditions between the mock and actual assessments, and explore the predictive factors in changes in students' health conditions via cohort study design. Thus, the findings can provide evidence about how the mock exam can affect or inform the students' physio-psychological state and assessment performance in a subsequent actual skills assessment. Fourth, it is possible that the Hawthorne effect [64] might have influenced the student's anxiety and stress levels due to observational and monitoring effects.

## Implications for future research and nursing education

This study offered some implications for future nursing education. First, the mock skill competency assessment showed that first-year undergraduate nursing students encountered both physiological stress and psychological anxiety when faced with mock assessment. Clinical nurses are usually exposed to high levels of stressors from emotions that arise in the context of patient care to the situations in which they practice. The stress experienced by undergraduate nursing students during their education can significantly influence their psychological attitudes as students and as healthcare professionals in the future. Our results highlighted the importance of considering assessment-related stress and anxiety in the context of student health. Therefore, wellness initiatives such as stress management and resilience promotion should be developed for health professional education programmes to address the emotional needs of students associated with stress and anxiety induced by assessments in the curriculum,

thus laying a good foundation for their mental health in the future [59]. Secondly, the test of HRV in our study indicates that HRV could be used as an index to evaluate the physiological adaptability of undergraduate nursing students to a changing environment of skill assessments or training involving changes in the autonomic nervous system. Measuring HRV with non-invasive devices also compensates well for the limitations of using self-reported questionnaires to measure perceived stress on students' academic performance. Understanding the relationship between HRV-measured stress and performance will help nursing educators develop counselling and educational programmes for stress management based on objective HRV results and performance. Thirdly, as the autonomic nervous system and the hypothalamic–pituitary–adrenal (HPA) axis are the two primary systems involved in adapting to stressful situations, HRV has been shown to be a powerful indicator of the autonomic nervous system, while salivary cortisol has been suggested as an indicator of HPA axis functioning. Further studies are suggested to explore the potential use of salivary cortisol as diagnostic tool for detecting stress-induced situations and investigate the relationship between HRV, stress response, and the cortisol level in student populations.

This study also provides implications for future studies. First, the current findings could only indicate the relationship between HRV and performance; that is, higher stress (lower HRV) could contribute to higher performance. However, whether those who performed better experienced higher stress before the assessment remains unclear. A two-way causal relationship could be further explored in future studies. Moreover, this study only investigated the students' overall mock skill competency assessment performance. Considering the mock skill competency assessment involved three aspects, students may perform diversely in different parts and show relevancy with HRV in the other perspective. Further studies may consider analysing the potential relevance between HRV and various aspects of the performance. Despite the correlations amongst perceived anxiety, HRV, and performance indicated, the mechanism for changes throughout assessment remains unclear. Given an optimal level of HRV is critical to the flexibility and resilience that characterises good health, and both higher and lower than normal levels may adversely affect cognitive processing and therefore academic performance. Therefore, to understand the role of HRV in clinical performance and facilitate optimal learning outcomes, it is necessary to explore the optimal levels of HRV that predict optimal nursing performance. Future qualitative studies may help enrich the understanding of the stressful experience during skill competency assessments and identify students' needs.

## Conclusion

This study investigated the perceived anxiety, stress levels (assessed by HRV) and mock skill competency assessment scores of first-year nursing students during a mock skill competency assessment. The results revealed significant changes in stress levels throughout the assessments, with higher stress levels (lower HRV) during the assessment. In addition, nursing students with higher stress levels (as measured by HRV) during and after assessments achieved higher scores, highlighting that certain degrees of stress can positively impact performance. Future studies are warranted to assess perceived anxiety and stress amongst diverse and representative samples in actual competency assessments and compare the changes in the students' conditions between the mock and real assessments.

## Supporting information

**S1 Fig. Pairwise comparisons of three time points for all participants and different gender.** Each node shows the sample average rank of HRV. (DOCX)

**S2 Fig. Pairwise comparisons of performance level during and 10 minutes after the assessment.** Each node shows the sample average rank of performance.
(DOCX)

**S3 Fig. Pairwise comparisons of three time points for different levels of performers.** Each node shows the sample average rank of performance.
(DOCX)

**S1 Table. Post-hoc comparison after the Kruskal-Wallis test for the HRV changes within gender groups across time.** Each row tests the null hypothesis that the Sample 1 and Sample 2 distributions are the same. Asymptotic significances (2-sided tests) are displayed. The significance level is 0.05.
(DOCX)

**S2 Table. Post-hoc comparison after the Kruskal-Wallis test for the HRV in three performance-level groups across time.** Each row tests the null hypothesis that the Sample 1 and Sample 2 distributions are the same. Asymptotic significances (2-sided tests) are displayed. The significance level is 0.05.
(DOCX)

**S3 Table. Post-hoc comparison after the Kruskal-Wallis test for the HRV changes within three performance-level groups across three time points.** Each row tests the null hypothesis that the Sample 1 and Sample 2 distributions are the same. Asymptotic significances (2-sided tests) are displayed. The significance level is 0.05.
(DOCX)

## Acknowledgments

We thank Well Being Digital Limited (WBD101) and Actywell Digital Limited (Hera Leto) for providing the earphone device and developing the mobile phone application for data collection. We would also like to express our thanks to two research assistants, Dr Mark Szeto and Ms Kenway Ng for data collection.

## Author Contributions

**Conceptualization:** Cho Lee Wong.

**Formal analysis:** Cho Lee Wong.

**Methodology:** Cho Lee Wong.

**Resources:** Mary Miu Yee Waye.

**Software:** Cho Lee Wong.

**Supervision:** Wai Tong Chien.

**Writing – original draft:** Cho Lee Wong.

**Writing – review & editing:** Cho Lee Wong, Wai Tong Chien, Mary Miu Yee Waye, Mark Wun Chung Szeto, Huiyuan Li.

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
