## [Decision Letter · Decision Letter 0]

24 Jan 2023

PONE-D-22-31368Nursing students’ perceived anxiety and heart rate variability in mock skill assessmentPLOS ONE

Dear Dr. Wong,

Thank you for submitting your manuscript to PLOS ONE. After careful consideration, we feel that it has merit but does not fully meet PLOS ONE’s publication criteria as it currently stands. Therefore, we invite you to submit a revised version of the manuscript that addresses the points raised during the review process.

Your paper has been assessed by three acknowledged experts in the field covered by the study. Overall, all the referees agree on the fact that the study has potential, but some key revisions are needed to reconsider my decision. Among their points, issues such as the description of the methods (needing various amendments and clarifications), presentation of results, and their subsequent interpretation stand out. Notwithstanding, please note that all the comments received must be responded to, providing suitable answers in a rebuttal letter, supporting your rationales with pertinent sources and arguments, and tracking all the changes made in the revised version of the manuscript.

We look forward to receiving your revised manuscript.

Kind regards,

Sergio A. Useche, Ph.D.

Academic Editor

PLOS ONE

Journal Requirements:

Reviewers' comments:

Reviewer's Responses to Questions

**Comments to the Author**

1. Is the manuscript technically sound, and do the data support the conclusions?

Reviewer #1: Yes

Reviewer #2: Yes

Reviewer #3: Yes

2. Has the statistical analysis been performed appropriately and rigorously? 

Reviewer #1: Yes

Reviewer #2: No

Reviewer #3: Yes

3. Have the authors made all data underlying the findings in their manuscript fully available?

Reviewer #1: No

Reviewer #2: Yes

Reviewer #3: No

4. Is the manuscript presented in an intelligible fashion and written in standard English?

Reviewer #1: Yes

Reviewer #2: Yes

Reviewer #3: Yes

5. Review Comments to the Author

Reviewer #1: This paper presents a study revolving around first year nursing students’ perceived and physilogically measured anxiety during a mock skill assessment.

I find the paper clearly written and well-structured. The authors do a good job of clearly defining the goals of the study early in the paper meanwhile the methods they employ are appropriate. However, I will list some points of discussion below, which I would appreciate if the authors considered before their study is published:

- Early in the paper, the authors state "Little is known about levels of stress and perceived anxiety and their relationship in the mock skill competency assessment". While this statement might be true in the current context, I believe that plenty is known regarding the relationship of stress, HRV, and assessed tasks in general. In fact, it is widely known that stress and task performance show an inverted U-shape relationship. I believe the authors should at least acknowledge the fact that a lot of studies have been conducted in the broad fields of psychology, physiology and skill assessment. The fact that a particular study has not validated this general knowledge in the specific domain of first year nursing students does not mean that the above principles do not apply or should necessarily be re-validated. The above being said, I believe that the present study lacks novelty and/or societal and scientific impact. I would like the authors to identify and emphasize what scientific gaps their study aims to fill.

- Please provide your working (or external) definitions of "stress" and "anxiety".

- I believe that an interesting part of this study (referring back to my first point) is the exploration of relationships between stress and performance. In this specific field, I would expect that a certain level of stress -which may be quite high- is expected in real-life nursing tasks. High-stake work in healthcare is objectively a stress-inducing domain. Therefore, would you agree that stress management and resilience are soft skills that nursing students should be developing? In simple words, is a certain level of stress during training/assessment not necessary to prepare students for their professional career? If yes, I encourage the authors to re-consider framing stress as not necessarily just as a possible negative influence on performance; but as an unavoidable consequence, or part of these students' potential work life that needs to be embraced and controlled.

- Line 199: please elaborate more on how these ranges were selected. It seems that medium-performing students occupy a significantly smaller range of scores. Also, it seems that no student failed this mock assessment; is that true? I would appreciate an overview of what percentage of the participants scored in which category.

- Line 280 and line 354: It seems that lower HRV (higher stress) is positively correlated with higher performance. However, the authors only address this observations from a single angle: higher stress causes higher academic performance. However, one may argue that the expectations of (generally)high-performing students can cause an increase in their stress measurements before and during an assessment. In simple words, students who expect themselves to do well in an assessment tend to have increased anxiety due to their own expectations. On the contrary, students who were not as well prepared, or simply do not care about their grades as much, showed lower stress for the respectively opposite reasons. I believe that this correlation is not necessarily a one-way causal relationship (as discussed in the paper) but a two-way relationship instead.

- I believe that the discussion section lacks a critical, well, discussion of the obtained results. Most of the discussion rotates around previous relevant studies and whether the results illustrated validate or contradict them. I encourage the authors to perform a fair, critical evaluation of their results. For example, the last paragraph of discussion is devoted to the relationship of gender and stress, however, this study showed that there is no such (significant) relationship. That being said, the discussion on this topic feels more like a literature review. In my opinion, it does not necessarily add any value to the study. Moreover, related to my previous point, the HRV-performance relationship seems to be borderline (statistically) insignificant, and could certainly be discussed in more detail. What is the p-value of both points' significance tests? A p-value of, for example, <=0.1 indicates a near-significant finding, which still deserves to be discussed.

- Minor points: a box appears in the .pdf version around line 165. Also, would you consider adding the tables to the main text instead of the appendix?

Reviewer #2: This paper addresses an interesting topic. Although the theoretical motivation for this paper is relatively sound, major revisions are necessary before publication. .

Below I highlight issues to be addressed:

Formatting issues need to be corrected (e.g. a square appears in particular places where a hyphen should be instead).

Thorough revision of the manuscript is necessary for correction of grammatical mistakes. Initially, the manuscript was relatively well written, but as it progressed, I came across more and more mistakes, especially in the resuts section (where there were also issues with reporting) and discussion. This is just one example: "The results indicated that high performers were physiological more stressful during and after the assessment. > physiologically more stressed.

p. 5. "If a person is under stress, the sympathetic activity will increase and at the same time, the vagal activity of the heart will decrease, resulting in an increase in heart rate (HR) and also a decrease in HRV (Traina et al., 2011). Several studies have found that when compared with non-academic occasions such as during the semester (Dimitriev et al., 2008), during holidays (Tharion et al., 2009) and after holidays (Melillo et al., 2011), university students exhibited lower HRV than before or during examintaton." >> If according to Traina et al., 2011, stress is associated with a decreased HRV, then the information in the second sentence does not follow. To clarify, you mean to say that HRV is lower during or before exams? If so, then the second sentence needs to be corrected.

Why is it relevant to mention that participants did not meet the WHO physical activity guideline? And the mean time of getting out of bed? Would not the average hours of sleep make more sense? Also, how do hours of sleep influence HR & HRV?

In terms of performance, since there were different parts to the examination, and perhaps students performed better in some than in others, this might lead to more insightful analyes and conclusions.

In the methods section, please further elaborate on how HRV/ RRI is calculated.

p. 14 -"The mean HRV as measured by RRI at 10 minutes before, during and 10 minutes after assessment were 688.73(ms) (SD = 116.98), 537.62(ms) (SD = 94.82) and 714.67(ms) 266 (SD =

267 119.68), respectively. " I am not sure I understand these values - what is "ms" an abbreviationf for? I am used to it representing milliseconds.

p. 15 You only report that there is a difference somewhere between the 3 times HRV was measured, but you do not report the output of the Friedman test nor post-hoc tests to demonstrate where the differences ere significant.

Remove the title for the section: "Different performers and their perceived anxiety and HRV" as it falls under the previous section already. Since you have values presented in tables, I would suggest just reffering to the table instead of repeating them in a text.

Since the assessment had 3 compontents, it might have been relevant to measure HRV spearately for those 3 components.

Since you are reporting results of non-parametric tests, you should be reporting medians and not means.

Use abbreviations/ terminology consistently. Sometimes you use RR interval, and sometimes you use RRI.

Discussion:

p. 18 "This study also showed that high-performing students had lower average HRV." Remove this sentence, since in the subsequent sentence you indicate that the difference was not significant.

The discussion should not include numbers/ statistics, but rather should be interpreting what was reported in the Results section.

I find it interesting that higher performance was correlated with lower HRV. This demonstrates that stress (as measured by lower HRV) seems to be beneficial. You discuss optimal levels of stress and differences between individuals in terms of their performance, yet it is not clear whether and how your conclusions are/can be supported by your data (refer to p. 19: "Therefore, students with stronger autonomic nervous system responses (i.e., greater reductions in HRV and parasympathetic activity in response to assessments), and, suggestively, greater autonomic flexibility, have been shown to be positively associated with higher performance (Mathewson et al., 2010)" > How can you make this conclusion based on your data / analyses?)

p. 20 "Nonetheless, our findings also highlighted a dynamic changing trajectory that three distinct groups of performers experience stress levels simultaneously, which may help predict the variation of physiological stress throughout the mock assessment in nursing freshmen and provide inspiration on psychological support during nursing education. " > You only compute the average HRV, so how are you highlighting a dynamic changing trajectory? What do you mean by "experience stress levels simultaneously?"? "and provide inspiration on psychological support" > this is vague

I would remove the discussion on gender differences, particularly since this was something that was not controlled for, nor theoretically motivated in the introduction.

Implications for further studies were rather shallow and could be discussed in a more critical manner.

Reviewer #3: Dear authors,

Thank you for this interesting manuscript. I can 100% imagine how these skill tests must be nerve wrecking.

Firstly, I strongly recommend to provide full statistical information, and not only p values.

Also, figures are much easier to interpret when you have data like this than providing numbers in texts of tables. E.g. line graphs with CI95% or SE bars.

If you say there are significant differences (e.g. page 15, line 284-285), also please tell the reader what the differences were (e.g its much better to say that Lisa is taller than Jenny, than to say the two girls are of different height). Again, figures make it much easier to then 'see' what that looks like.

page 16, section Correlations amongst perceived anxiety, HRV, and mock skill competency assessment score: You don't have to write out the whole correlation table....

Page 17, line 313, you mean self-reported anxiety?

What is meant on page 21 at the following alinea? IF it wasn't significant, there was no difference? And if I remember right there are hardly any gender differences found right?

399 Interestingly, female students experienced higher levels of perceived anxiety after the

400 assessment compared to their pre-assessment scores, despite statistically insignificance was

401 reported.

I don't agree with the implications for further research. Your study did not show " the importance of the mock assessment" and also, how does it show that anxiety should be considered? If anything, your study suggests that yes, ofcourse like always, assessments cause some anxiety, but they don't cause problems right?

Oerall: the discussion can be more concise.

ALso, there are some small errors (often missing a word or e.g. missing an 'a' in front of a word, for example line 325 "previous study showed" instead of "A previous study" and "students may not be sure what to be expected" or "The difference likely be explained" (I am not going to note all mistakes, a careful editing will take care of them). Overall the writing is clear though.

6. PLOS authors have the option to publish the peer review history of their article (what does this mean?). If published, this will include your full peer review and any attached files.

Reviewer #1: No

Reviewer #2: No

Reviewer #3: No

---

## [Author Response · Author response to Decision Letter 0]

22 Mar 2023

Response to reviewers

We thank the Editor and Reviewers for their time and consideration. We have modified the manuscript in response to the comments. For your convenience, the line and page numbers referenced below refer to the corresponding line and page numbers in the revised manuscript. The changes are highlighted in red in the manuscript, except the part on changes resulting from editing.

Response to Editor:

Thank you very much for your comments. Our answers are as follows:

Response to Editor comment No. 1: Thanks for your comments. We carefully checked the Title, Author, Affiliations Formatting Guidelines of PLoS One, adjusted the formatting issues and ensured the title page met the journal’s style requirements. Please refer to page 1.

In addition, we also carefully checked the PLoS One style template for main body and formatted the whole text to make the manuscript meet the journal’s instructions, including the file naming. Tables were included directly after the paragraph in which they were first cited. Please refer to the whole text.

2. We note that you have indicated that data from this study are available upon request. PLOS only allows data to be available upon request if there are legal or ethical restrictions on sharing data publicly. For more information on unacceptable data access restrictions, please see http://journals.plos.org/plosone/s/data-availability#loc-unacceptable-data-access-restrictions

Response to Editor comment No. 2a: Thanks for your comments. The detail of the reason that data were available upon request due to ethical restrictions was added in ‘Availability of data and materials’ section. We also provided the contact emails of the university's clinical research ethics committee and corresponding author for sending data requests. 

The datasets generated and analysed during the current study are not publicly available due to privacy protection and ethical restrictions by the Joint CUHK-NETEC Clinical Research Ethics Committee. The Chinese University of Hong Kong’s Research Ethics Board has imposed data sharing restrictions as they do not allow identifiable or de-identified data to be electronically transmitted outside of university including open access repositories. Requests can be sent to Clinical Research Ethics Office (Phone: 852 3505 3935; email: crec@cuhk.edu.hk). Requests can also be sent to the corresponding author (jojowong@cuhk.edu.hk).

Response to Editor comment No. 2b: Thanks for your comments. Our data availability has ethics restrictions. The detail of the reason that data were available upon request due to ethical restrictions was added in ‘Availability of data and materials’ section. We also provided the contact emails of the university's clinical research ethics committee and corresponding author for sending data requests. 

Response to Editor comment No. 3: Thanks for your comments. We agree that the ethics statement should only appear in the Methods section and thus deleted the ethics statement at the end of the manuscript. Please refer to page 33, lines 601-607.

Response to Editor comment No. 4: Thanks for your comments. We uploaded a file named ‘STROBE_checklist_cross-sectional.docx’ as a Supporting Information file when we first submitted our manuscript in the system. As it is a reporting guideline checklist for a cross-sectional study for review purpose only by the editorial office, not for publication. Thus, we did not add an in-text citation and the caption for this Supporting Information file at the end of our manuscript.

On the other hand, as we further did other statistical analyses according to the reviewers’ comments, the related findings were added as Supporting Information files. The captions of our Supporting Information files were added at the end of the manuscript, and the in-text citations were also updated accordingly. Please refer to pages 17-19, and 45-46.

Response to Reviewer #1:

This paper presents a study revolving around first year nursing students’ perceived and physilogically measured anxiety during a mock skill assessment.

I find the paper clearly written and well-structured. The authors do a good job of clearly defining the goals of the study early in the paper meanwhile the methods they employ are appropriate. However, I will list some points of discussion below, which I would appreciate if the authors considered before their study is published:

Response: Thank you very much for your thorough review. We have answered each of your points below.

1. Early in the paper, the authors state "Little is known about levels of stress and perceived anxiety and their relationship in the mock skill competency assessment". While this statement might be true in the current context, I believe that plenty is known regarding the relationship of stress, HRV, and assessed tasks in general. In fact, it is widely known that stress and task performance show an inverted U-shape relationship. I believe the authors should at least acknowledge the fact that a lot of studies have been conducted in the broad fields of psychology, physiology and skill assessment. The fact that a particular study has not validated this general knowledge in the specific domain of first year nursing students does not mean that the above principles do not apply or should necessarily be re-validated. The above being said, I believe that the present study lacks novelty and/or societal and scientific impact. I would like the authors to identify and emphasize what scientific gaps their study aims to fill.

Response to Reviewer comment No. 1: Thank you for your comment. We agree that a particular study has not validated this general knowledge in the specific domain of first-year nursing students does not mean that the above principles do not apply or should necessarily be re-validated. We added the related description and acknowledged in the background part that a lot of studies have been conducted to indicate the relationship between stress and performance in the broad fields of psychology, physiology, and skill assessment. Please refer to page 7, lines 130-133.

In addition, we also highlighted the educational and scientific impacts of this study. 

First, evaluating HRV is an objective way to know about the stress level in undergraduate nursing students during the mock skill competency assessment. As most studies adopted scales/questionnaires related to anxiety to assess students’ perceived anxiety, adopting HRV to objectively reflect nursing students’ stress may help promote the application of physiological measurement technologies in the nursing education field.

Second, few data exist on the levels of stress in undergraduate nursing students in health programmes using subjective and physiological measures and their relationship to performance in the mock skill assessment. Monitoring physiological HRV-based stress in undergraduate nursing students before the actual assessment may help build evidence on new ways to evaluate students’ performance and professional overall qualities in nursing programmes.

We emphasised this significance in the introduction part. Please refer to pages 4-5, lines 80-83, pages 6-7, lines 114-118, pages 7-8, lines 137-139.

2. Please provide your working (or external) definitions of "stress" and "anxiety".

Response to Reviewer comment No. 2: Thanks for your comments. We added the definitions of ‘stress’ and ‘anxiety’ in the Introduction part. Please refer to page 4, lines 62–66. 

3. I believe that an interesting part of this study (referring back to my first point) is the exploration of relationships between stress and performance. In this specific field, I would expect that a certain level of stress -which may be quite high- is expected in real-life nursing tasks. High-stake work in healthcare is objectively a stress-inducing domain. Therefore, would you agree that stress management and resilience are soft skills that nursing students should be developing? In simple words, is a certain level of stress during training/assessment not necessary to prepare students for their professional career? If yes, I encourage the authors to re-consider framing stress as not necessarily just as a possible negative influence on performance; but as an unavoidable consequence, or part of these students' potential work life that needs to be embraced and controlled.

Response to Reviewer comment No. 3: Thanks for your comment. We do agree that extensive demands are often placed on nurses in their actual workplaces that can lead to high rates of stress and burnout, which thus have a vast array of negative consequences for the patient, the health care organization, the nursing profession, and the mental and physical health of the nurses as an individual. There is no doubt that stress management and resilience facilitation may assist nurses and nursing students to manage the demands of the healthcare environment and to build resilience. However, although a certain level of stress is unavoidable during assessments/trainings, if the stressors cannot be properly adjusted or controlled, it is likely to affect the subsequent enthusiasm for the nursing profession, job satisfaction in the future practice, and physical and mental health in the process of career development. That is why we aimed to investigate physiological and psychological stress and its relationship with performance at the very beginning of their undergraduate nursing study. This is also one of the important implications of our findings on nursing education. Evaluating physiological and psychological stress may form the basis on which the decision was made to incorporate stress management and resilience facilitation into nursing programmes to assist students to manage the stressors of undergraduate nursing study and future clinical practice., thus promoting occupational mental health. 

4. Line 199: please elaborate more on how these ranges were selected. It seems that medium-performing students occupy a significantly smaller range of scores. Also, it seems that no student failed this mock assessment; is that true? I would appreciate an overview of what percentage of the participants scored in which category.

Response to Reviewer comment No. 4: Thank you for your comment. The levels and score ranges are identified by the programme director, course coordinator and education committee of the school. It has been widely used in various nursing courses in the researchers’ nursing school and as an evaluation standard for nursing students’ performance. We added the details in the Methods part. Please refer to page 12, lines 218-220.

Moreover, it’s true that medium-performing students occupy a significantly smaller range of scores, and no student failed this mock assessment in our study. Based on students' knowledge mastery and performance in previous years, their mock assessments were generally at an average level. The real meaning of the mock assessment is to let students understand their current mastery degree and thus improve their shortcomings accordingly before the actual assessment.

The percentages of the participants in each category could be found in Table 1. Please refer to page 16. Students with high, medium, and low performance accounted for 22.22% (n = 20), 48.89% (n = 44), and 28.29% (n = 26), respectively. We also added one sentence to point out the overview of the results to avoid repeating the results in the table. Please refer to page 18, lines 326-327.

5. Line 280 and line 354: It seems that lower HRV (higher stress) is positively correlated with higher performance. However, the authors only address this observations from a single angle: higher stress causes higher academic performance. However, one may argue that the expectations of (generally)high-performing students can cause an increase in their stress measurements before and during an assessment. In simple words, students who expect themselves to do well in an assessment tend to have increased anxiety due to their own expectations. On the contrary, students who were not as well prepared, or simply do not care about their grades as much, showed lower stress for the respectively opposite reasons. I believe that this correlation is not necessarily a one-way causal relationship (as discussed in the paper) but a two-way relationship instead.

Response to Reviewer comment No. 5: Thanks for your comments. We do agree that a two-way causal relationship between HRV and performance may exist. However, our current research data could only reflect the one-way relationship, that is higher stress is positively correlated with higher performance. The two-way relationship deserves further exploration in larger-sample studies. We added this as a further research direction in the implications part. Please refer to page 30, lines 551-555.

6. I believe that the discussion section lacks a critical, well, discussion of the obtained results. Most of the discussion rotates around previous relevant studies and whether the results illustrated validate or contradict them. I encourage the authors to perform a fair, critical evaluation of their results. For example, the last paragraph of discussion is devoted to the relationship of gender and stress, however, this study showed that there is no such (significant) relationship. That being said, the discussion on this topic feels more like a literature review. In my opinion, it does not necessarily add any value to the study. Moreover, related to my previous point, the HRV-performance relationship seems to be borderline (statistically) insignificant, and could certainly be discussed in more detail. What is the p-value of both points' significance tests? A p-value of, for example, <=0.1 indicates a near-significant finding, which still deserves to be discussed.

Response to Reviewer comment No. 6: Thanks for your comments. We rewrote the discussion part according to our findings more critically. Please refer to pages 21-31. 

In addition, we defined the statistical significance of all tests as a p-value < 0.05 (two-sided) in our study, which could be found on page 15, lines 276-277. A p-value greater than 0.05 means that deviation from the null hypothesis is not statistically significant, and the null hypothesis is not rejected. We agree that the possible reasons for the insignificant differences could be discussed critically according to the research data. Therefore, we discussed more on the HRV-performance relationships. Please refer to pages 23-24, lines 422-423, 427-430, 440-444. 

With regard to the discussion on gender and stress, as one of our important objectives was to examine the differences in perceived anxiety, HRV and performance between female and male students across the mock skill competency assessment. Previous studies also indicate gender is a potential influencing factor on students’ stress and performance. We also did the analysis, although no significant differences were found between male and female students. However, interesting findings were that female students experienced lower levels of perceived anxiety and HRV-measured stress than male students before and after the assessment, but showed higher levels of HRV-measured stress than male students during the assessment, and the mean score of the female perceived anxiety after the assessment was a little bit higher than those before the assessment, although these results were insignificant. We rephrased the statements on the gender difference and did some critical analysis for the non-significant results. More studies are warranted to further investigate the gender difference on this topic, which may provide insights into nursing education based on gender. Please refer to pages 26-27, lines 484-507.

7. Minor points: a box appears in the .pdf version around line 165. Also, would you consider adding the tables to the main text instead of the appendix?

Response to Reviewer comment No. 7: Thanks for your comments. We carefully checked the MS Word document of the manuscript and ensured that no other box appeared around line 165. Please refer to page 9.

In addition, we added the tables directly after the paragraph in which they were first cited rather than at the end of the manuscript. Please refer to pages 16-21.

Response to Reviewer #2:

This paper addresses an interesting topic. Although the theoretical motivation for this paper is relatively sound, major revisions are necessary before publication. 

Below I highlight issues to be addressed:

Response: Thank you very much for your thorough review. Our answers are as follows:

1. Formatting issues need to be corrected (e.g. a square appears in particular places where a hyphen should be instead).

Response to Reviewer comment No. 1: Thank you for comment. We carefully checked the formatting issues throughout the Word document and used a standard hyphen, ensuring no square appearing. 

2. Thorough revision of the manuscript is necessary for correction of grammatical mistakes. Initially, the manuscript was relatively well written, but as it progressed, I came across more and more mistakes, especially in the results section (where there were also issues with reporting) and discussion. This is just one example: "The results indicated that high performers were physiological more stressful during and after the assessment. > physiologically more stressed.

Response to Reviewer comment No. 2: Thank you for your comment. We apologise for the poor language of our manuscript. The revised manuscript was sent for professional editing to improve the use of English, especially the results and discussion parts. We really hope that the flow and language have been substantially improved. Please refer to the whole text.

3. p. 5. "If a person is under stress, the sympathetic activity will increase and at the same time, the vagal activity of the heart will decrease, resulting in an increase in heart rate (HR) and also a decrease in HRV (Traina et al., 2011). Several studies have found that when compared with non-academic occasions such as during the semester (Dimitriev et al., 2008), during holidays (Tharion et al., 2009) and after holidays (Melillo et al., 2011), university students exhibited lower HRV than before or during examintaton." >> If according to Traina et al., 2011, stress is associated with a decreased HRV, then the information in the second sentence does not follow. To clarify, you mean to say that HRV is lower during or before exams? If so, then the second sentence needs to be corrected.

Response to Reviewer comment No. 3: Thank you for your comment. We apologise for the confusion by the previous version of the manuscript. Yes, we agree that the second sentence in the text did not follow the first sentence from Traina’s research. We corrected the description of the second sentence accordingly. Please refer to page 6, line 107.

4. Why is it relevant to mention that participants did not meet the WHO physical activity guideline? And the mean time of getting out of bed? Would not the average hours of sleep make more sense? Also, how do hours of sleep influence HR & HRV?

Response to Reviewer comment No. 4: Thank you for your comment. According to WHO physical activity guideline, adults aged 16-64 years should do at least 150–300 minutes of moderate-intensity aerobic physical activity throughout the week. Physical activity has significant health benefits for hearts, bodies and minds, reduces symptoms of depression and anxiety, enhances thinking, learning, and judgment skills, and improves overall well-being. The lack of adequate intensity of physical exercise also partly indicated that nursing undergraduates are experiencing physiological and psychological stress. As the average time of sleep was not the research focus of this research. As such, we delete this information in the results part to avoid misleading. Please refer to page 15, line 284-285. We may consider exploring the influencing factors of HRV by regression analysis in further studies. 

5. In terms of performance, since there were different parts to the examination, and perhaps students performed better in some than in others, this might lead to more insightful analyes and conclusions.

Response to Reviewer comment No. 5: Thank you for your comment. Although the final assessment included a 2-hour written examination and a skill competency examination. The performance in this study was only referred to the mock skill competency assessment scores. In addition, although the mock skill competency examination contained three aspects: assessments and planning skills, implementation skills, and evaluation skills on performing a wound dressing, investigating the overall performance in the mock skill competency assessment was our research objective, not the performance of other dimensions of the examination. We indeed believe that students may perform diversely in different parts of the examination. This provides us with a further research direction to investigate and compare the students’ performance and HRV in different parts of the examination, which will enrich the findings and conclusions. We added this implication in the text. Please refer to page 30, lines 555-560. 

6. In the methods section, please further elaborate on how HRV/ RRI is calculated.

Response to Reviewer comment No. 6: Thank you for your comment. We further added the details of the data generation and analysis of HRV/RR interval in Methods part. When students were asked to wear the earphones, the HRV could be directly recorded using the ActivHeartsTM dynamic heart rate sensing technology in the earphone device according to the guidelines of the task force of the European Society of Cardiology and the North American Society of Pacing and Electrophysiology. Please refer to page 11, lines 200–203.

7. p. 14 -"The mean HRV as measured by RRI at 10 minutes before, during and 10 minutes after assessment were 688.73(ms) (SD = 116.98), 537.62(ms) (SD = 94.82) and 714.67(ms) 266 (SD = 267 119.68), respectively. " I am not sure I understand these values - what is "ms" an abbreviationf for? I am used to it representing milliseconds.

Response to Reviewer comment No. 7: Thank you for your comment. Yes, ‘ms’ is an abbreviation for ‘milliseconds’, which is the unit of RR interval. We wrote out the full term when ‘ms’ was firstly used. Please refer to page 14, line 268.

8. p. 15 You only report that there is a difference somewhere between the 3 times HRV was measured, but you do not report the output of the Friedman test nor post-hoc tests to demonstrate where the differences are significant.

Response to Reviewer comment No. 8: Thanks for your comment. We agree that the Kruskal Wallis Test and the Friedman text were only the first steps to indicating the significant differences amongst the three groups. We indeed should add the post hoc test to further identify the differences. The current results could not show more detail whether any two groups in the three groups show any differences. Thus, we first added the chi-square values of the Friedman test, and we further did the post hoc multiple pairwise comparisons and added the results of significant differences between any two groups. Please refer to pages 17-18, lines 318-323, pages 18-19, lines 338-345, and supporting information (S1-3 Table, S1-3 Fig).

9. Remove the title for the section: "Different performers and their perceived anxiety and HRV" as it falls under the previous section already. Since you have values presented in tables, I would suggest just reffering to the table instead of repeating them in a text.

Response to Reviewer comment No. 9: Thanks for your comments. We agree that the contents below the subheading ‘Different performers and theirs perceived anxiety and HRV’ belong to the previous section. Thus, we deleted this subheading. Please refer to page 18, line 329.

Moreover, we agree that Table 2 could well represent the results of HRV in different three levels of performers at three-time points. We shortened the description by referring the Table 2 rather than repeating the results. Please refer to page 18, lines 330-335.

10. Since the assessment had 3 components, it might have been relevant to measure HRV separately for those 3 components.

Response to Reviewer comment No. 10: Thank you for your comment. Although the mock skill competency examination contained three aspects: assessments and planning skills, implementation skills, and evaluation skills on performing a wound dressing, investigating the overall performance in the mock skill competency assessment was our research objective. We indeed believe that students may perform diversely in different parts of the examination. This provides us with a further research direction to investigate and compare the students’ performance and HRV in different parts of the examination, which will enrich the findings and conclusions. We added this implication in the text. Please refer to page 30, lines 555-560. 

11. Since you are reporting results of non-parametric tests, you should be reporting medians and not means.

Response to Reviewer comment No. 11: Thanks for your comment. We revised and reported the medians (P25, P75) for anxiety scores, HRV, and performance. Please refer to pages 16-19.

12. Use abbreviations/ terminology consistently. nishihSometimes you use RR interval, and sometimes you use RRI.

Response to Reviewer comment No. 12: Thanks for your comments. We apologise for the confusion generated by the previous version of the manuscript. We checked all the abbreviations throughout the whole text and used them consistently. Specifically, we replaced ‘RRI’ with ‘RR interval’ in the text to make them consistently used. 

13. Discussion: p. 18 "This study also showed that high-performing students had lower average HRV." Remove this sentence, since in the subsequent sentence you indicate that the difference was not significant.

Response to Reviewer comment No. 13: Thanks for your comments. We deleted this sentence. We agree that the subsequent sentence could represent our research finding. Please refer to Page 23, line 418. 

14. The discussion should not include numbers/ statistics, but rather should be interpreting what was reported in the Results section.

Response to Reviewer comment No. 14: Thanks for your comments. We agree that the discussion should not include numbers/statistics or repeat much information on the results part. We deleted the statistics in the Discussion, avoided restatement of the results, and pay more attention to interpreting the possible reasons for the research findings. Please refer to pages 21-28.

15. I find it interesting that higher performance was correlated with lower HRV. This demonstrates that stress (as measured by lower HRV) seems to be beneficial. You discuss optimal levels of stress and differences between individuals in terms of their performance, yet it is not clear whether and how your conclusions are/can be supported by your data (refer to p. 19: "Therefore, students with stronger autonomic nervous system responses (i.e., greater reductions in HRV and parasympathetic activity in response to assessments), and, suggestively, greater autonomic flexibility, have been shown to be positively associated with higher performance (Mathewson et al., 2010)" > How can you make this conclusion based on your data / analyses?)

Response to Reviewer comment No. 15: Thanks for your comments. We apologise for the confusion of your reading. Research from Mathewson et al. 2010 could be a shred of evidence to help explain why students with lower HRV had higher performance in our study. We did not make this conclusion but refer to this as a possible reason to support our finding that is aligned with the finding from Yoo et al., 2021. We revised the related statements and added a detailed discussion. Please refer to page 24, lines 440 to 447.

16. p. 20 "Nonetheless, our findings also highlighted a dynamic changing trajectory that three distinct groups of performers experience stress levels simultaneously, which may help predict the variation of physiological stress throughout the mock assessment in nursing freshmen and provide inspiration on psychological support during nursing education. " > You only compute the average HRV, so how are you highlighting a dynamic changing trajectory? What do you mean by "experience stress levels simultaneously?"? "and provide inspiration on psychological support" > this is vague

Response to Reviewer comment No. 16: Thanks for your comments. As we measured the average level of HRV at three time points: 10 minutes before, during, and 10 minutes after the assessment, we would like to reflect a dynamic changing process throughout the mock skill assessment process at the three time points. Certainly, fewer time points limited us showing a more concrete changing process. Adding more time points is suggested in further studies to reflect a concrete dynamic process. Please refer to page 23, line 432.

What we meant to say ‘experience stress levels simultaneously’ referred to that high, medium, and low performers showed a changing curve of HRV, respectively. We revised the related interpretation in the text. Please refer to page 25, lines 458-459.

In addition, we further added more details on ‘provide inspiration on psychological support’. Please refer to page 25, lines 461-463.

17. I would remove the discussion on gender differences, particularly since this was something that was not controlled for, nor theoretically motivated in the introduction.

Response to Reviewer comment No. 17: Thanks for your comments. As one of our important objectives was to examine the differences in perceived anxiety, HRV and performance between female and male students across the mock skill competency assessment. Previous studies also indicate gender is a potential influencing factor on students’ stress and performance. We also did the analysis, although no significant differences were found between male and female students. Thus, we reserved and revised the discussion on gender differences in our study to analyse possible reasons. Please refer to pages 26-28, lines 484 to 507. We also added more descriptions of the supportive evidence and potential mechanism of gender differences in stress and performance in the background part. Please refer to page 5, lines 94-96, and page 6, lines 111-113. 

18. Implications for further studies were rather shallow and could be discussed in a more critical manner.

Response to Reviewer comment No. 18: Thank you for your comments. We rewrote the implications part critically. Please refer to pages 29-31. 

Response to Reviewer #3:

Dear authors,

Thank you for this interesting manuscript. I can 100% imagine how these skill tests must be nerve wrecking.

Response: Thank you very much for your thorough review. Our answers to each question are as follow.

1. Firstly, I strongly recommend to provide full statistical information, and not only p values. Also, figures are much easier to interpret when you have data like this than providing numbers in texts of tables. E.g. line graphs with CI95% or SE bars. If you say there are significant differences (e.g. page 15, line 284-285), also please tell the reader what the differences were (e.g its much better to say that Lisa is taller than Jenny, than to say the two girls are of different height). Again, figures make it much easier to then 'see' what that looks like.

Response to Reviewer comment No. 1: Thank you for your comment. We checked carefully and added the Z values of the Mann-Whitney U test and Wilcoxon Signed Rank test and the chi-square value of the Friedman test to ensure provide full statistical information.

In addition, we do agree that figures are much easier to interpret the research findings. As the data in our study were not normally distributed, the interquartile range (IQR) may be better to reflect their probability distribution than 95%CI. In this case, box plots were added to compare the changes of the perceived anxiety and HRV in males and females and in three groups of performers throughout the mock skill assessment process. Please refer to Figs 1 and 2. 

Moreover, we agree that the statements of the significant differences were not unclear. We rephrased the statements for the significant differences. For the HRV of the high, medium and low performers, as the Kruskal-Wallis Test was the first step to see whether the three groups showed significant differences. If we would like to see whether any two groups amongst the three groups show significant differences, a further step is needed – selecting all pairs for multiple pairwise comparisons. To make the results show more details, we added the results of the post hoc test after Kruskal-Wallis Test to further indicated any significant differences between the two groups. Please refer to pages 17-18, lines 318-323, pages 18-19, lines 338-345, and supporting information (S1-3 Table, S1-3 Fig).

2. page 16, section Correlations amongst perceived anxiety, HRV, and mock skill competency assessment score: You don't have to write out the whole correlation table....

Response to Reviewer comment No. 2: Thank you for your comment. We agree that the text should not repeat all the results in the table. We shortened the summary of the correlation results and highlighted the key points of Table 3 in the text. Please refer to pages 20-21, lines 364-379.

3. Page 17, line 313, you mean self-reported anxiety?

Response to Reviewer comment No. 3: Thank you for your comment. We agree that ‘subjective anxiety’ means ‘self-reported anxiety’/‘perceived anxiety’. We revised the ‘subject anxiety’ into ‘perceived anxiety’ in the text. Please refer to page 21, line 388.

4. What is meant on page 21 at the following alinea? IF it wasn't significant, there was no 4 difference? And if I remember right there are hardly any gender differences found right?

399 Interestingly, female students experienced higher levels of perceived anxiety after the

400 assessment compared to their pre-assessment scores, despite statistically insignificance was

401 reported.

Response to Reviewer comment No. 4: Thank you for your comments. We apologise for the confusion generated by the previous version of the manuscript. Yes, no significant differences across gender were found in the scores of the perceived anxiety and HRV before and after the assessment. However, interesting findings were that female students experienced lower levels of perceived anxiety and HRV-measured stress than male students before and after the assessment, but showed higher levels of HRV-measured stress than male students during the assessment, and the mean score of the female perceived anxiety after the assessment was a little bit higher than those before the assessment. Although these results were insignificant, we would like to discuss the nuances, suggesting further exploration in future studies. We rephrased the statements on the gender difference and did some critical analysis for the non-significant results. Please refer to pages 26-28, lines 484-507.

5. I don't agree with the implications for further research. Your study did not show " the importance of the mock assessment" and also, how does it show that anxiety should be considered? If anything, your study suggests that yes, ofcourse like always, assessments cause some anxiety, but they don't cause problems right?

Response to Reviewer comment No. 5: Thank you for your comments. Yes, we agree that our research findings did not mainly highlight the importance of the mock assessment and anxiety issue. Instead, our findings would like to indicate the importance of HRV-measured stress in undergraduate nursing students and the acceptability of HRV measurement by a wearable earphone device using ActivHeartsTM dynamic heart rate sensing technology in undergraduate nursing students. This could be applied in future nursing research and provides insights into stress management for undergraduate nursing students in nursing education programmes. We rewrote the implications for future research and practice. Please refer to pages 29-31. 

6. Oerall: the discussion can be more concise.

ALso, there are some small errors (often missing a word or e.g. missing an 'a' in front of a word, for example line 325 "previous study showed" instead of "A previous study" and "students may not be sure what to be expected" or "The difference likely be explained" (I am not going to note all mistakes, a careful editing will take care of them). Overall the writing is clear though.

Response to Reviewer comment No. 6: Thanks for your comments. We revised the discussion part and sincerely hope that the discussion part is now better, please refer to pages 20-31.

We apologise for the poor language of our manuscript. We have now checked the whole text and corrected the small errors. The revised manuscript was sent for professional editing to improve the use of English. We really hope that the flow and language level have been substantially improved.

---

## [Decision Letter · Decision Letter 1]

30 May 2023

PONE-D-22-31368R1Nursing students’ perceived anxiety and heart rate variability in mock skill competency assessmentPLOS ONE

Dear Dr. Wong,

Thank you for submitting your manuscript to PLOS ONE. After careful consideration, we feel that it has merit but does not fully meet PLOS ONE’s publication criteria as it currently stands. Therefore, we invite you to submit a revised version of the manuscript that addresses the points raised during the review process. Although the paper has been substantially improved by you, there are some issues still needing your attention. Below, you will find a set of comments (most of them relatively simple) raised by our Reviewer # 2 during their second review of the manuscript. Please make sure to address all of them with all rigor possible, and to provide satisfactory responses to all the points delivered.

We look forward to receiving your revised manuscript.

Kind regards,

Sergio A. Useche, Ph.D.

Academic Editor

PLOS ONE

Journal Requirements:

Reviewers' comments:

Reviewer's Responses to Questions

**Comments to the Author**

1. If the authors have adequately addressed your comments raised in a previous round of review and you feel that this manuscript is now acceptable for publication, you may indicate that here to bypass the “Comments to the Author” section, enter your conflict of interest statement in the “Confidential to Editor” section, and submit your "Accept" recommendation.

Reviewer #2: All comments have been addressed

2. Is the manuscript technically sound, and do the data support the conclusions?

Reviewer #2: Partly

3. Has the statistical analysis been performed appropriately and rigorously? 

Reviewer #2: Yes

4. Have the authors made all data underlying the findings in their manuscript fully available?

Reviewer #2: Yes

5. Is the manuscript presented in an intelligible fashion and written in standard English?

Reviewer #2: No

6. Review Comments to the Author

Reviewer #2: The authors have considerably improved the manuscript, but there are still some points to be addressed:

Line 39: space missing (10min)

Line 62: “Stress is a situation”. Revisit this definition – stress is not a situation. Definitions of stress and anxiety could be further elaborated upon/ contextualized.

Line 94: “Additionally, evidence suggests that female students are more likely to focus their perceived stress on academic performance than male students [21].” This sentence is unclear. Are they more likely to attribute their stress to academic performance? Please clarify. In your response letter, you state gender differences to be a primary goal of your study – yet you do not discuss it much in depth in the introduction. What might be reasons for this gender difference? In what other domains can a gender difference be found. Also – are you referring to gender or sex differences? I presume the latter?

Line 97: “Alternatively” seems to be the incorrect connector. Alternatively to what?

On page 6, you motivate the importance of this study in terms of incorporating stress measurement as a part of the assessment. What would be ethical implications of this? Would knowing that stress levels are being measured also possibly lead to increased anxiety?

Line 137: “Monitoring physiological HRV-based stress in undergraduate nursing 8 138 students before the actual assessment may help build evidence on new ways to evaluate 139 students’ performance and overall professional qualities in nursing programmes.” In line with my previous comment, what are the ethical considerations of this statement?

Line 305: I believe mean should be medians? The same goes for line 375. Wherever the statistics used were nonparametric, you should be reporting medians. Otherwise, Means and SDs.

Formatting: (p > 0.05) (Table 2) >> (p > 0.05, Table 2)

In the discussion, 2nd parapgrah, you discuss the fact that there were no differences between anticipatory stress and stress during/ after the assessment. No differences were present between different levels of performers and anticipatory stress. Would the fact that this was a mock exam, with no real consequences, have led to these null results? I would be interested if there is any more literature out there on anticipatory stress, and possibly interactions with trait anxiety, for example.

When you say that previous research has yielded inconsistent findings, reiterate how they have been inconsistent ( rather than referring to studies).

Line 440 – “Testing HRV helps confirm the autonomic nervous system activity and sympathetic nervous system/parasympathetic nervous system balance.” This sentence is unclear. What do you mean by confirm?

Line 445: “Therefore, students with stronger 445 autonomic nervous system responses (i.e., more significant reductions in HRV and 446 parasympathetic activity in response to assessments), and, suggestively, greater autonomic 447 flexibility, have a positive effect on higher performance [56].”

Line 447: future studies

Line 551 – “First, the current findings 552 could only indicate the one-way possible causal relationship between HRV and performance; 553 that is, higher stress (lower HRV) could contribute to higher performance.” If you only tested for differences in groups, how can the relationship be causal?

Line 559 – “Further studies may dig data into analysing” – dig data?

A clarification/ discussion of what “optimal” stress levels is necessary.

In terms of the scientific and societal motivation of this study, I find the view presented is at times to narrow. I think it is important how these findings might generalize elsewhere. Instead of focusing so much on how this is relevant for nursing assesment, I think it is also important to motivate the importance of your study from a more fundamental perspective.

With regards to my point concerning findings pertaining to gender in the discussion – I still suggest you remove any speculation, especially considering your findings were insignificant.

7. PLOS authors have the option to publish the peer review history of their article (what does this mean?). If published, this will include your full peer review and any attached files.

Reviewer #2: No

---

## [Author Response · Author response to Decision Letter 1]

28 Jun 2023

Response to reviewers

We thank the Editor and Reviewers for their time and consideration. We have modified the manuscript in response to the comments. For your convenience, the line and page numbers referenced below refer to the corresponding line and page numbers in the revised manuscript. The changes are highlighted in red in the manuscript, except for the part on changes resulting from editing.

Response to Editor:

Thank you very much for your comments. Our answers are as follows:

Response to Editor comment No. 1: Thanks for your comments. We carefully checked the reference list and ensured that no retracted papers were cited. The reference list is complete.

Response to Reviewer #2:

The authors have considerably improved the manuscript, but there are still some points to be addressed:

Response: Thank you very much for your thorough review. Our responses are as follows:

1. Line 39: space missing (10min)

Response to Reviewer comment No. 1: We have now added the space between ‘10’ and ‘minutes’. Please refer to page 2, line 38.

2. Line 62: “Stress is a situation”. Revisit this definition – stress is not a situation. Definitions of stress and anxiety could be further elaborated upon/ contextualized.

Response to Reviewer comment No. 2: We have now revised the definition of stress. Further elaboration on students’ stress and anxiety was also added. Please refer to page 4, lines 61-70.

3. Line 94: “Additionally, evidence suggests that female students are more likely to focus their perceived stress on academic performance than male students [21].” This sentence is unclear. Are they more likely to attribute their stress to academic performance? Please clarify. In your response letter, you state gender differences to be a primary goal of your study – yet you do not discuss it much in depth in the introduction. What might be reasons for this gender difference? In what other domains can a gender difference be found. Also – are you referring to gender or sex differences? I presume the latter?

Response to Reviewer comment No. 3: We have now revised this sentence. Please refer to page 6, line 99.

As our secondary objective was to investigate the differences in perceived anxiety, HRV and performance between female and male students across (before, during and after) the mock skill competency assessment, we have now added more interpretations on sex differences. Please refer to page 6, lines 100-104.

Besides, we referred to sex differences in our study rather than gender differences. We have now revised the description. Please refer to page 7, line 121.

4. Line 97: “Alternatively” seems to be the incorrect connector. Alternatively to what?

Response to Reviewer comment No. 4: We have now revised the description. Please refer to page 6, line 105. 

5. On page 6, you motivate the importance of this study in terms of incorporating stress measurement as a part of the assessment. What would be ethical implications of this? Would knowing that stress levels are being measured also possibly lead to increased anxiety?

Response to Reviewer comment No. 5: We agreed that it might lead to increased anxiety if students know that stress levels are being measured, especially when self-reported questionnaires are heavily relied on for measuring stress. As such, we attempted to analyse and interpret the changes in the cognitive and psychological states of students during the examination process by measuring the physio-psychological reactions. Physio-psychological response is an objective indicator, which can explain the state of the object under investigation at various levels without undue interference or influence on learning/examination. As an index to evaluate the physiological adaptability of the participants to a changing environment in skill assessment, the test of heart rate variability is recognised as useful and scientific in confirming the autonomic nervous system activity and balance level in objectively evaluating the psychological and emotional states of an individual [1]. In addition, students were required to arrive at the assessment venue 30 minutes before the assessment. They rested for at least 20 minutes before the measure, breathing spontaneously without talking. 

Nonetheless, we have added the limitation and ethical implications due to the observational and monitoring effects of this study. Please refer to page 29, lines 527-528. “Fourth, it is possible that the Hawthorne effect [2] might have influenced the student’s anxiety and stress levels due to observational and monitoring effects”.

1. Yoo HH, Yune SJ, Im SJ, Kam BS, Lee SY. Heart Rate Variability-Measured Stress and Academic Achievement in Medical Students. Med Princ Pract. 2021;30(2):193-200. doi: 10.1159/000513781.

2. Sedgwick P, Greenwood N. Understanding the Hawthorne effect. BMJ. 2015;351:h4672. doi: 10.1136/bmj.h4672.

6. Line 137: “Monitoring physiological HRV-based stress in undergraduate nursing 138 students before the actual assessment may help build evidence on new ways to evaluate 139 students’ performance and overall professional qualities in nursing programmes.” In line with my previous comment, what are the ethical considerations of this statement?

Response to Reviewer comment No. 6: Thank you for noting this. We have now added the limitation and ethical implications due to the observational and monitoring effects of this study. Please refer to page 29, lines 527-528. 

7. Line 305: I believe mean should be medians? The same goes for line 375. Wherever the statistics used were nonparametric, you should be reporting medians. Otherwise, Means and SDs.

Response to Reviewer comment No. 7: We have now changed ‘mean’ to ‘median’ in which nonparametric tests were used for data not normally distributed. Please refer to page 17, lines 311 and 313, page 18, line 335, and page 20 lines 368 and 371, page 21, line 373 and 376.

8. Formatting: (p > 0.05) (Table 2) >> (p > 0.05, Table 2)

Response to Reviewer comment No. 8: Thanks for noting this. We carefully checked the formatting issue and revised it accordingly. Please refer to page 17, lines 318-319, and page 20, line 362. 

9. In the discussion, 2nd paragraph, you discuss the fact that there were no differences between anticipatory stress and stress during/ after the assessment. No differences were present between different levels of performers and anticipatory stress. Would the fact that this was a mock exam, with no real consequences, have led to these null results? I would be interested if there is any more literature out there on anticipatory stress, and possibly interactions with trait anxiety, for example.

Response to Reviewer comment No. 9: Thank you for highlighting this. We do discuss that no significant differences were reported 10 minutes before the assessment, however, significant differences were shown between the three groups during and 10 minutes after the assessment. Concerning the similar levels of anticipatory stress, we can find that the mock skill assessment stress exists and there are significant differences between the three groups during and 10 minutes after the assessment. We added further discussion about the potential reasons for the non-significant anticipatory stress between the three groups before the mock skill assessment. Please refer to page 23, lines 419-423.

10. When you say that previous research has yielded inconsistent findings, reiterate how they have been inconsistent (rather than referring to studies).

Response to Reviewer comment No. 10: Thank you for your comment. We have now added specific descriptions of the inconsistent findings from previous studies. Please refer to pages 24, lines 427-431. 

11. Line 440 – “Testing HRV helps confirm the autonomic nervous system activity and sympathetic nervous system/parasympathetic nervous system balance.” This sentence is unclear. What do you mean by confirm?

Response to Reviewer comment No. 11: We have now further elaborated on this point. Please refer to pages 24-25, lines 443-453.

12. Line 445: “Therefore, students with stronger 445 autonomic nervous system responses (i.e., more significant reductions in HRV and 446 parasympathetic activity in response to assessments), and, suggestively, greater autonomic 447 flexibility, have a positive effect on higher performance [56].”

Response to Reviewer comment No. 12: Thanks for your comments. We further revised this sentence to make it clearer and coherent within the context. Hope the revised version with further elaborations can ease your concern. Please refer to pages 24-25, lines 446-453.

13. Line 447: future studies

Response to Reviewer comment No. 13: We have now revised the description. Please refer to page 25, line 453. 

14. Line 551 – “First, the current findings 552 could only indicate the one-way possible causal relationship between HRV and performance; 553 that is, higher stress (lower HRV) could contribute to higher performance.” If you only tested for differences in groups, how can the relationship be causal?

Response to Reviewer comment No. 14: We have now deleted ‘causal’ for the relationship between HRV and performance. Please refer to page 30, line 557.

15. Line 559 – “Further studies may dig data into analysing” – dig data?

Response to Reviewer comment No. 15: We have now revised the description. Please refer to page 31, line 564.

16. A clarification/ discussion of what “optimal” stress levels is necessary.

Response to Reviewer comment No. 16: Thanks for your comments. A clarification of an optimal stress level was added. Please refer to page 31, lines 567-570. We also highlighted the significance of an optimal level of stress in the Background. Please refer to page 7, lines 123-127.

17. In terms of the scientific and societal motivation of this study, I find the view presented is at times too narrow. I think it is important how these findings might generalize elsewhere. Instead of focusing so much on how this is relevant for nursing assessment, I think it is also important to motivate the importance of your study from a more fundamental perspective.

Response to Reviewer comment No. 17: We do agree that highlighting the importance of this study is needed, not only from the perspective of highlighting the importance of HRV measurement applied in nursing curriculums. Hence, we not only highlighted the importance of developing wellness initiatives (e.g. stress management and resilience promotion) in health professional education programme (page 29, lines 536-541), but we also added implications to help understand the mechanism of stress response in student population. Please refer to page 30, lines 549-555. 

18. With regards to my point concerning findings pertaining to gender in the discussion – I still suggest you remove any speculation, especially considering your findings were insignificant.

Response to Reviewer comment No. 18: Thank you for highlighting this. We have now deleted the discussion on non-significant sex differences in perceived anxiety and HRV before and after the assessment.

---

## [Editor Report · Decision Letter 2]

16 Oct 2023

Nursing students’ perceived anxiety and heart rate variability in mock skill competency assessment

PONE-D-22-31368R2

Dear Dr. Wong,

We’re pleased to inform you that your manuscript has been judged scientifically suitable for publication and will be formally accepted for publication once it meets all outstanding technical requirements.

Kind regards,

Sergio A. Useche, Ph.D.

Academic Editor

PLOS ONE

Additional Editor Comments (optional):

The remaining changes made by the authors are OK. I can accept the paper in its current status.
---

## [Editor Report · Acceptance letter]

18 Oct 2023

PONE-D-22-31368R2 

Nursing students’ perceived anxiety and heart rate variability in mock skill competency assessment 

Dear Dr. Wong:

I'm pleased to inform you that your manuscript has been deemed suitable for publication in PLOS ONE. Congratulations! Your manuscript is now with our production department. 

Kind regards, 

on behalf of

Dr. Sergio A. Useche 

Academic Editor

PLOS ONE